# Systematic Review and Meta-Analysis on MS-Based Proteomics Applied to Human Peripheral Fluids to Assess Potential Biomarkers of Bipolar Disorder

**DOI:** 10.3390/ijms23105460

**Published:** 2022-05-13

**Authors:** Joao E. Rodrigues, Ana Martinho, Vítor Santos, Catia Santa, Nuno Madeira, Maria J. Martins, Carlos N. Pato, Antonio Macedo, Bruno Manadas

**Affiliations:** 1CNC—Center for Neuroscience and Cell Biology, University of Coimbra, 3004-504 Coimbra, Portugal; joao.e.a.rodrigues@gmail.com (J.E.R.); anajmartinho@gmail.com (A.M.); catiajmsanta@gmail.com (C.S.); martins.mjrv@gmail.com (M.J.M.); 2CIBB—Centre for Innovative Biomedicine and Biotechnology, University of Coimbra, 3004-504 Coimbra, Portugal; vitorsantos74@gmail.com; 3Faculty of Medicine, University of Coimbra, 3004-504 Coimbra, Portugal; nunogmadeira@gmail.com; 4Psychiatry Department, Centro Hospitalar e Universitário de Coimbra, 3004-561 Coimbra, Portugal; 5CIBIT—Coimbra Institute for Biomedical Imaging and Translational Research, University of Coimbra, 3000-548 Coimbra, Portugal; 6Medical Services, University of Coimbra Medical Services, 3004-517 Coimbra, Portugal; 7Department of Psychiatry and Behavioral Sciences, SUNY Downstate Health Sciences University, Brooklyn, NY 11203, USA; carlos.pato@downstate.edu; 8III Institute for Interdisciplinary Research, University of Coimbra (IIIUC), 3030-789 Coimbra, Portugal

**Keywords:** proteomics, mass spectrometry, bipolar disorder, biomarkers, human peripheral fluids

## Abstract

Bipolar disorder (BD) is a clinically heterogeneous condition, presenting a complex underlying etiopathogenesis that is not sufficiently characterized. Without molecular biomarkers being used in the clinical environment, several large screen proteomics studies have been conducted to provide valuable molecular information. Mass spectrometry (MS)-based techniques can be a powerful tool for the identification of disease biomarkers, improving prediction and diagnosis ability. Here, we evaluate the efficacy of MS proteomics applied to human peripheral fluids to assess BD biomarkers and identify relevant networks of biological pathways. Following PRISMA guidelines, we searched for studies using MS proteomics to identify proteomic differences between BD patients and healthy controls (PROSPERO database: CRD42021264955). Fourteen articles fulfilled the inclusion criteria, allowing the identification of 266 differentially expressed proteins. Gene ontology analysis identified complement and coagulation cascades, lipid and cholesterol metabolism, and focal adhesion as the main enriched biological pathways. A meta-analysis was performed for apolipoproteins (A-I, C-III, and E); however, no significant differences were found. Although the proven ability of MS proteomics to characterize BD, there are several confounding factors contributing to the heterogeneity of the findings. In the future, we encourage the scientific community to use broader samples and validation cohorts, integrating omics with bioinformatics tools towards providing a comprehensive understanding of proteome alterations, seeking biomarkers of BD, and contributing to individualized prognosis and stratification strategies, besides aiding in the differential diagnosis.

## 1. Introduction

Bipolar disorder (BD) is a complex affective disorder integrating a group of clinical conditions characterized by biphasic episodes of mania or hypomania and depression expressed as recurrent episodic changes in mood, energy levels, thought, and behavior interpolated with periods of euthymia. BD includes bipolar I disorder, defined by the presence of a manic episode, and bipolar II disorder, defined by the presence of a hypomanic episode and a major depressive episode [1,2,3].

The estimated lifetime prevalence is 0.6% for bipolar I disorder, 0.4% for bipolar II disorder, and 2.4% for the broader spectrum of bipolar disorders [4], explaining 0.4% of total disability-adjusted life years (DALYs) and 1.3% of total years lived with disability (YLDs) in 2013 [5]. BD usually begins in late adolescence or early adulthood, with around 75% of the cases having their first episode at that age period [6]. This early-onset and the associated level of disability mean that BD is the 4th leading cause of global disease burden in adolescents and young adults [7], disrupting the attainment of relationships and educational and occupational milestones. In addition, as bipolar disorders affect the economically active population, high costs to society are incurred in terms of direct healthcare costs and the costs of disability [2].

A high prevalence of psychiatric and medical comorbidities is common in affected individuals. Bipolar disorders are comorbid with other psychiatric disorders, including anxiety disorders, substance use disorders, attention-deficit/hyperactivity disorders, and personality disorders. This comorbidity makes diagnosing and managing bipolar disorders more difficult and is associated with poorer outcomes [1,2,3]. Cardiovascular disorders, diabetes, and obesity are highly comorbid with BD and arise earlier in the life course compared with the general population [8,9]. The mortality gap between populations with BD and the general population is principally a result of excess deaths from cardiovascular disease and suicide, with a loss of approximately 10–20 potential years of life [3].

Mental health services are increasingly adopting strategies to diagnose and treat BD as early as possible. The delay in diagnosis, as well as inappropriate treatment, can result in repeated mood episodes, persistent subthreshold symptoms, development of co-morbidities, and progression of the disease with cognitive impairment, functional decline [10,11], and an excess of premature mortality [12]. The high complexity, heterogeneity, and absence of biologically relevant diagnostic markers of BD are some of the factors that contribute to the misdiagnosis of the illness [13].

Currently, the diagnosis of BD is made by a comprehensive clinical assessment with the administration of standardized diagnostic instruments, supplemented when possible with third-party information (e.g., family members, and clinical records). The identification of the various diagnostic categories that are part of the bipolar spectrum is made based on several operational criteria specified in international classification systems, such as the International Classification of Diseases, 11th revision (ICD-11) [14], and the Diagnostic and Statistical Manual of Mental Disorders, 5th edition (DSM-5) [15]. Additionally, clinical strategies applied to identify individuals at risk for BD remain mostly descriptive and have insufficient predictive validity, with only a minority of the individuals “at-risk” making the transition to BD [16,17]. Given the progressive nature of BD, the delay in the correct diagnosis, and the risk of exposure to inappropriate pharmacological and psychosocial treatments, obtaining a timely and accurate diagnosis is extremely important. Thus, the identification of reliable biomarkers is becoming imperative.

Despite the extensive efforts to elucidate BD causes and mechanisms, these are still elusive [18]. It is understood that the etiology and pathophysiology of BD involve several factors, including genetic, neurochemical, and environmental factors [18,19]. Moreover, pathways underlying the neuroprogression in BD have been associated with immune dysfunction [20,21], mitochondrial dysfunction [22], oxidative stress [23], neurotransmitter systems [24,25], and impairment of cellular resilience and neural plasticity [26]. In addition, emotional factors (e.g., maladaptive emotional processing and deficient emotion regulation) have been suggested as core factors in different psychopathologies, including bipolar disorders [27].

To improve the knowledge about these complex disorders, omics approaches have arisen as relevant methodologies to achieve comprehensive information on disease pathogenesis and support the identification of reliable strategies for disease prediction and diagnoses [28,29], towards the ultimate goal of improving patient care and outcome. However, the translation from research to successful clinical omics-based tests is still far from acceptable compared to the potential of these approaches.

Biomarker discovery, development, and application have been the center of extensive interest, especially with the recent emergence of new technologies such as proteomics-based approaches [13]. Proteomics is considered a powerful tool in the omics field, examining the proteins expressed in a cell, tissue, or organism, enabling real-time evaluation of an individual state, health versus disease, and potentially predicting the susceptibility to develop a specific disorder [30,31]. Extensive efforts have already been made toward the identification of biomarkers of several psychiatric disorders such as schizophrenia, depression, and BD [32,33,34]. In mental health disorders, a biomarker, or a panel of biomarkers, may have several purposes: (i) to correctly diagnose and stratify a psychiatric patient in a field where several diseases may have overlapping clinical symptoms; (ii) to better classify at-risk individuals; (iii) to perform prognosis; (iv) to be used as therapeutic monitoring; and (v) to be used as predictive of therapy compliance [17,35].

Proteomics strategies can generate several distinct levels of information, particularly (i) identification of proteins in a sample at a given moment; (ii) expression levels of proteins or quantitative proteomics; (iii) identification or quantification of post-translational modifications (such as phosphorylation, glycosylation, and acetylation, among others) of those proteins; (iv) determination of protein–protein interactions; and (v) proteomic functional studies [13,36,37].

The increasing use of proteomics is closely related to the technological advances in mass spectrometry (MS), optimization in sample preparation, and computer data processing ability to deal with the large amount of information generated by the MS-based technologies [37]. The success of MS in proteomics relies on its high specificity and sensitivity, mainly due to advances in liquid chromatography-tandem MS (LC-MS/MS), giving answers to different purposes and questions [38].

Although in the beginning, MS-proteomics strategies were focused on screening approaches to qualitatively characterize proteins in complex matrices, over the last years, quantitative proteomics has become the analysis of choice when comparing proteomes, given that most of the biological changes of interest are slight differences in the amount of a protein present in a given situation and not an abrupt change testifying its presence or absence [39]. Nowadays, proteomics tools usually combine qualitative and quantitative strategies, either using targeted or untargeted techniques to identify and quantify proteins in complex biological matrices [40,41].

Numerous studies have employed MS-based techniques in BD, with most of the studies being initially performed in post-mortem brain tissue [42,43,44]. Although very informative, brain tissue has some drawbacks, such as the susceptibility of confounding factors, namely age, chronicity of the disease and medication, and the fact that the tissue is static with no possibility of being manipulated, disturbed, or having longitudinal samplings [45,46]. Although the search for biomarkers in psychiatric disorders started with brain tissue and cerebrospinal fluid samples, the whole body concept has emerged with great success based on the understanding that the brain and a variety of physiological conditions are reflected in the contents of body fluids [47,48]. This link, created between the brain and the periphery, enhanced the search for biomarkers in body fluids that could be easily available, for example, in the blood (plasma, serum, and peripheral blood mononuclear cells—PBMCs) [32,49,50], urine [51], saliva [52], and even tears [53]. In this way, current proteomics studies have focused on peripheral fluids by integrating MS-based techniques towards a more comprehensive picture of BD concerning its onset, progression, and response to medication.

This article provides a systematic review and meta-analysis on the use of MS-based methods in proteomic studies to assess biomarkers or a panel of biomarkers associated with bipolar disorder (BD) based only on the analysis of peripheral fluids.

## 2. Method

As our study used systematic review and meta-analysis strategies, ethical approval of this study and an informed consent statement is not required. We included all articles that met all the keywords that specified the objective of the study. This systematic review followed a methodological protocol based on the PRISMA Statement, which was registered in the PROSPERO database (identifier: CRD42021264955).

### 2.1. Search Strategy

Research manuscripts included in the systematic review were identified through a computer-based search conducted in two independent databases: PUBMED and Web of Science (WoS). The search was conducted in all fields using the following keywords: BIPOLAR DISORDER, AND MASS SPECTROMETRY, AND PROTEOMIC*, until December 2020. The PUBMED and WoS databases were last searched on 29 June 2021. Moreover, references in all relevant studies were screened for research papers that might have been missed during the database searches. Two authors, J.R. and A.M., assessed the eligibility of the studies for inclusion, and disagreements were discussed with a third author, B.M. Extracted data were entered into a computerized spreadsheet for analysis. Then the reference lists of the included studies, excluded studies, and previous reviews were searched. When necessary, the authors of some studies were contacted to request additional information.

### 2.2. Eligibility Criteria

Articles were included if they met the following criteria: (a) research design included the use of mass spectrometry-based techniques for proteome profiling and/or quantification; (b) research performed in human peripheral fluids samples, collected with minimally-invasive or non-invasive sampling procedures (which resulted in the exclusion of CSF samples as their collection in many countries is not a standard procedure for psychiatric disorders); (c) research design included a group of identified BD patients and a control group comprising healthy controls, and (d) a peer-reviewed English language journal.

### 2.3. Data Extraction

Two authors, J.R. and A.M., independently extracted the following data from the eligible studies, according to a pre-specified protocol of data extraction: (1) authors; (2) year of publication; (3) participants’ characteristics (including diagnosis type of BD, sample size and group comparison, mean age, mean illness duration, gender, medication status, type of peripheral samples, and clinical criteria applied); (4) analytical technique; (5) sample preparation (protein depletion or/and enrichment); (6) differences between protein levels of BD patients as measured against controls or other mental disorders; and (7) altered pathways (Table 1 and Table 2).

Any discrepancies between the extracted data were resolved in a group meeting.

### 2.4. Quality of Evidence

The quality of the studies was determined using the QUADOMICS methodology criteria (Appendix A), and it was evaluated independently by two authors (Appendix A). QUADOMICS is an adaptation of QUADAS—a quality evaluation tool for use in systematic reviews of diagnostic accuracy studies, which takes into account the technical particularity presented by omics methodologies [54,55].

### 2.5. Statistical and Gene Ontology Analysis

To perform the meta-analyses, the effect size for each measured protein was standardized to log2 Fold Change. In this way, effect sizes and corresponding significance that were heterogeneously expressed in the studies as (i) ratio or log(ratio) and the corresponding *p*-value, or (ii) group averages and the corresponding standard deviations were all transformed into fold change and corresponding *p*-values. Proteins in which it was possible to compute the effect size in at least two research studies were included in the meta-analysis.

A forest plot was created to present the output data, being the conventional way to report meta-analysis results. Meta-analysis was performed in R version 4.0.3 combined with Rstudio, using the following R packages: “meta” [56], “metafor” [57], and “dmetar” [58]. Gene ontology analysis was performed using MetaboAnalyst 5.0 [59] and the KEGG Mapper Color tool [60].

## 3. Results

### 3.1. Study Selection and Characteristics of Included Articles

The search strategy followed for selecting the eligible studies included in our systematic review/meta-analysis is summarized in the flow chart shown in Figure 1, following PRISMA 2020 [61]. A total of 116 potentially relevant manuscripts were identified from PubMed and WoS database searches, and twelve additional works were found from other sources. Based on the abstracts’ review, 65 manuscripts met the eligibility criteria, which were assessed for inclusion. Of these, 22 were identified as reviews, seven studies were performed in mice/rats, four studies were performed in cell lines, and 18 studies analyzed brain tissue. In total, 14 studies met all eligibility criteria and were included in this systematic review/meta-analysis.

All manuscripts included in this review were published from 2010 onward. The basic characteristics of the eligible studies are presented in Table 1 and are detailed below.

In the last five years, an increasing interest in the subject has been observed, with nine out of the 14 publications being from this period (see Appendix A). In fact, since 2014, two or more articles per year have been published using MS proteomics strategies to study BD (except in 2016, without any article published), whereas prior to 2014, only three studies were found [50,62,63].

**Table 1 ijms-23-05460-t001:** Demographic summary of all the studies included in the systematic review of bipolar disorder and biomarkers discovery using MS-based method in human peripheral fluids.

First Author	Year	Bipolar Disorder (BD)				Controls			Other Disorders (OD)				Clinical Criteria	Ref.
		*n*	Age	Illness Duration	Gender (m/f)	*n*	Age	Gender (m/f)	*n*	Age	Illness Duration	Gender (m/f)		
L. Smirnova	2019	23	32 (21–52)	8 (5–11)	14/9	24	28 (21–55)	6/18	33 (SCZ)	34 (28–40)	7 (4–16)	11/22	ICD-10	[64]
G.S. Pessoa	2019	19	41 ± 17	6.4 ± 6.1	7/12	13	38 ± 16	3/10	19 (SCZ)	37 ± 11	7.6 ± 5.4	13/6	ICD-10	[65]
Y.H. Cheng	2018	57	(18–50)	2.2 (0.25–12)	27/30	94	(18–50)		---	---	---	---	ICD-10	[66]
B. Petrov	2018	12	14 ± 2.0	---	---	13	14 ± 2.4		11 (MDD)	14 ± 1.2	---	---	K-SADS-PL-W	[67]
C. Knochel	2017	25	38 ± 10	8.9 ± 5.5	19/6	93	34 ± 11	44/39	29 (SCZ)	37 ± 11	12 ± 7.8	21/8	DSM-IV	[33]
J.R. De Jesus	2017	14	36 ± 9.0	4.5 ± 4.3	5/9	12 (3 HCF; 9 HCNF)	39 ± 9 (HCF); 35 ± 8 (HCNF)	1/2 (HCF); 2/7 (HCNF)	23 (SCZ); 4 (OD)	34 ± 9 (SCZ); 31 ± 5 (OD)	8.7 ± 7.5 (SCZ); 4.5 ± 2.9 (OD)	17/6 (SCZ); 3/1 (OD)	ICD-10	[32]
J.J. Ren	2017	30	28 ± 7.0	15.1 ± 20.4 weeks (depressive episode)	[65] 17/13	30	28 ± 6.0	15/15	30 (MDD)	30 ± 4.9	19.9 ± 25.7 weeks (depressive episode)	16/14	DSM-IV	[68]
Y.R. Song	2015	45 BD I (10 euth; 20 dep.; 15 man)	28 ± 9.5 (euth) 27 ± 9.1 (dep) 29 ± 8.0 (man)	---	4/6 (eut); 8/12 (dep); 6/9 (man)	20	28 ± 5.0	8/12	---	---	---	---	DSM-IV-Axis I	[49]
J. Chen	2015	20 (BD II)	---	---	---	30	---	---	30 (MDD)	---	---		DSM-IV-Axis I	[69]
L. Giusti	2014	15	41 ± 9.3	13 ± 9.6	4/11	15	39 ± 12	10/5	11 (MDE)	37 ± 9.4	9.5 ± 7.1	2/9	DSM-IV	[70]
J. Iavarone	2014	17	---	---	---	31	---		32 (SCZ)	---	---	---	DSM-IV	[52]
M. Herberth	2011	32 (BD I/II: 16/16) 16 PBMCs (BD I/II: 8/8)	34 ± 10 (serum) 36 ± 9.0 (PBMCs)	9.9 ± 8.6 (serum); 12 ± 8.6 (PBMCs)	13/19 (serum); 6/10 (PBMCs)	32 serum; 15 PBMCs	33 ± 6.6 (serum); 33 ± 7.3 (PBMCs)	13/19 (serum); 6/9 (PBMCs)	---	---	---	---	DSM-IV	[50]
A. Sussulini	2011	15 BD + Li; 10 BD − Li (euth)	40 ± 13 (+Li); 42 ± 17 (-Li)	1–28 (+Li); 1–20 (−Li)	6/9 (+Li); 3/7 (−Li)	15	31 ± 15	6/9	---	---	---	---	---	[63]
A. Sussulini	2010	BD + Li = 15; BD − Li = 10 (euth)	---	---	---	25	---	---	---	---	---	---	---	[62]

BD I, bipolar disorder type I; BD II, bipolar disorder type II; euth, euthymic; dep, depressive; man: maniac; HCF, familiar healthy control; HCNF, non-familiar healthy control; BD + Li, BD patients treated with lithium; BD − Li, BD patients treated with other drugs; SCZ: schizophrenia; OD, other disorders; MDD, major depressive disorder; MDE, major depressive episode.

### 3.2. Number of Patients

The number of BD patients included in the studies showed a strong heterogeneity, varying between 12 [67] to 57 [66] BD patients, with a median value of 25 patients (Table 1 and Appendix A). Comparing BD vs. control, the number of patients is well balanced, except for two studies in which the number of control individuals is 3-fold higher than BD patients [33,66]. Similarly, for BD vs. other disorders, a well-balanced number of subjects between groups was observed, except for one study [52].

### 3.3. Diagnostic Criteria

The majority of the selected studies applied the Diagnostic and Statistical Manual for Mental Disorders fourth edition (DSM-IV) criteria [33,49,50,52,68,69,70] (7 studies), followed by four studies applying the International Classification of Diseases 10th edition (ICD-10) criteria [32,64,65,66], and only one study applied the Kiddie Schedule for Affective Disorders and Schizophrenia for School-Age Children-Present and Lifetime Episode (K-SADS-PL-W) clinical criteria [67]. Interestingly, all studies using ICD-10 criteria were published after 2017. However, there were two studies [62,63], both prior to 2012, in which no information about diagnostic criteria was given.

### 3.4. Age

BD patients average age in the studies ranged from 14 [67] to 42 years [63], with the majority of the studies having average values between 28 and 42 years (9 studies) and one study only mentioning the cohort age range [66]. One study [67] was focused on young BD patients (an average of 14 years). Three studies did not provide information about the cohort’s age [52,62,69], being all these studies prior to 2015. Overall, the average age of the used cohort was similar between each group in each study (BD vs. control vs. other disorders).

### 3.5. Illness Duration

The cohort average illness duration varied between 2 [66] and 13 [70] years, with six studies having average values above 4.5 years [32,33,50,64,65,70]. Ren et al. [68] described the duration of the depressive episode. There is a higher number of studies (six studies) with a lack of information related to the illness duration [49,52,62,63,67,69], and five of these studies were published before 2016 [49,52,62,63,69].

### 3.6. Gender

The gender information in the BD group was heterogeneous. A higher number of female patients was observed in 6 studies [32,49,50,63,65,70], two with more male patients [33,64], two publications well-balanced between genders [66,68], and four with no information related to the gender ratio [52,62,67,69].

Consistently, in the control group, six publications had a higher number of females [32,49,50,63,64,65], two studies had a balanced number of samples [33,68], and only one study had more male participants [70]. In six studies, no gender information was given (two more studies than in the BD group) [50,52,62,66,67,69].

For other disorders (*n* = 9), three studies had a higher number of males [32,33,65], two with more females [64,70], one was well balanced [68], and three studies did not have any information [52,67,69].

Comparing gender information between BD vs. control, the gender is well balanced in both groups in one study [68], groups are differently balanced in three studies [33,64,70], a higher number of females in both groups in four studies [32,49,63,65], and there are six studies in which no gender information is given in at least one of the groups [50,52,62,66,67,69].

Overall, there are three studies published before 2015 [52,62,69] that only mentioned the number of BD patients. The remaining cohort information is lacking, namely age, illness duration, and gender.

All studies had a clinical control group (*n* = 14). Moreover, five studies also compared BD patients with SCZ [32,33,52,64,65]; four studies with major depressive disorder (MDD) or major depressive episode (MDE) [67,68,69,70]; and one study with other disorders [32]. In this systematic search, the studies assessing the effects of psychotropic drugs in BD therapeutics (*n* = 3) were also highlighted [32,62,63].

Table 2 summarizes the information related to the cohort information, diagnostic criteria, biological sample and type of sampling, drug naïve or under therapy, type of MS-based method, other techniques applied, use of depletion or enrichment strategies, differentially expressed proteins identified, and altered pathways and major findings.

### 3.7. Type of Sample and Sampling

The most prevalent biological sample was serum with eight studies, followed by plasma with five studies and PBMCs with three studies. Only one study was performed using saliva [52]. The trend of studying serum and plasma has been dominant in the last five years, with four studies being performed in serum [32,64,65,67], four in plasma [33,49,68,69], and one where both serum and plasma were assessed [66] (see Appendix A).

As mentioned above, plasma and serum represent important biological materials for disease diagnosis and disease profiling, widely used for proteomics-based biomarker discovery. However, the comprehensive analysis of these blood components is challenging due to their complexity and wide dynamic protein concentration range. In fact, high abundant proteins present a barrier to detecting and measuring potentially relevant proteins that are usually present in lower concentrations (medium and low abundance proteins). 

Several strategies have been used to overcome this wide dynamic range, particularly the depletion of high abundant proteins and enrichment of low abundant proteins of interest. Considering the selected studies (Table 2), there is a higher use of depletion strategies in the last five years than in the previous period, with four studies [32,49,64,68] out of nine studies in total. A particular study performed a comprehensive analysis of different depletion approaches towards the selection of the best performing method, suggesting the use of a commercial ProteoMiner kit as the best strategy for removing proteins of high abundance [32]. In the last five years, only one study applied labeling protocols, iTRAQ, using a prior depletion method [68].

No clear trend was observed for the analysis of individual vs. pooled samples. In fact, overall, eight studies used pooled samples, whereas six studies used individual samples. Considering the last three years, the ratio is similar to four studies using pooled samples [32,65,67,68] against three studies with individual samples [33,64,66].

### 3.8. Drug Naïve or Minimally Medicated 

There has been interest in performing studies with drug naïve/minimally medicated BD patients in the last three years. In fact, from 2010 to 2015, no study selected drug naïve patients, whereas between 2017 and 2019, three out of seven studies had patients minimally medicated [64,66,68]. 

### 3.9. MS-Based Methods

Examining the selected studies between 2010 and 2015, the use of MS-based methods was mainly for identification purposes, with the use of MALDI-TOF-MS being prevalent, with four studies applying this technique [49,62,63,69]. In fact, of the seven studies before 2015, only two used MS-based methods (LC-MS/MS) for quantification purposes [50,52]. In fact, one of the studies applied an MS method to quantify proteins in PBMCs samples but multiplexed immunoassays for the measurement of proteins in serum samples [50]. 

This trend was reversed in the last three years, with six studies out of seven using MS strategies for protein (semi-)quantitative measurements [33,64,65,66,67,68]. The increasing application of MS methods towards the quantitative measurement of proteins reflects remarkable developments/advances in MS-based platforms, mainly in technical issues and statistical tools dealing with BIG DATA towards a more comprehensive overview of the proteome. During this period, LC-MS/MS analysis was the prevalent MS-based technique, being used in six studies.

Two studies applied ICP-MS-based methods to assess the interactions between metals and proteins [62,65]. The use of ICP-MS-based methods served to evaluate the ionomic profile of the BD patients and increase the knowledge related to the evaluation of BD and the effects of conventional drug treatment from a metallomics point of view.

### 3.10. Other Techniques

Other techniques have also been applied for quantitative measurements with two main strategies: untargeted and targeted approaches. 

For untargeted approaches, two-dimensional gel electrophoresis (2DE), as 2D-Difference Gel Electrophoresis (2D-DIGE), was the most used analytical technique, mainly between 2010–2015 [49,50,62,63,69]; 2DE methods are widely used to separate and quantify macromolecules, such as proteins [71]. The ability of 2DE to obtain information on thousands of protein isoforms based on the physical parameters of isoelectric point and molecular weight in a single analysis is a strong advantage of this technique. However, 2DE has as its main limitation the poor reproducibility (low sensitivity and narrow linear dynamic ranges), although this has been reduced with the use of multiplexing methods, such as fluorescent 2D-DIGE.

Magnetic resonance imaging (MRI) was also used in one study [33] to detect and underline morphological changes occurring in BD and schizophrenia patients.

For targeted approaches, immunoassay methods have been applied to validate specific proteins identified as differentially expressed. The most used immunoassays methods applied to protein’s expression pattern validation were enzyme-linked immunosorbent assay (ELISA) [64,66,67,69] and Western blot (WB) [49,50,67,70], with four studies each. Interestingly, while WB had a strong use in the studies before 2015 [49,50,70], after 2017, the immunoassay of choice for validation was ELISA [64,66,67]. Another immunoassay method applied in one study included immunoturbidimetric analysis [63]. Moreover, flow cytometry was also used in one study for the analysis of PBMCs [66].

**Table 2 ijms-23-05460-t002:** Proteomic studies of bipolar disorder and biomarkers discovery using MS-based method in human peripheral fluids. The proteins identified as altered are represented by their accession number as described in UniProt (the corresponding protein name and entry name are described in Appendix A).

Author (year)	Cohort Information	Sample	Type of Sampling	DRUG NAIVE	MS-Based Method	Other Techniques	Quantification Method	Depletion/Enrichment	Altered Proteins	Altered Pathways	Ref.
L. Smirnova (2019)	23 BD; 33 SCZ; 24 CTR	Serum	Individual	Yes	LC-MS/MS	ELISA (Q6UB98; P33151)	MS	Yes/No	**BD vs. CT vs. SCZ:****↑** (O15417; O95445; P02666; P02745; P02753; P05090; P05452; P07360; P13671; P15924; P17948; P23141; P33151; P46013; Q01538; Q86YZ3; Q9HCI5; Q9UBP9); **↓** (A8K2U0; O75820; O95347; P00748; P01011; P02649; P02750; P05154; P11532; P22792; P42684; P60709; P63261; P78527; P81605; P84098; P98164; Q08380; Q15811; Q16610; Q5H9R4; Q6UB98; Q7Z478; Q8TE73; Q96BK5; Q96KN2; Q9UGM5)	**BD:** immune response, regulating transport processes across the cell membrane and cell communication, development of neurons and oligodendrocytes, and cell growth. **SCZ:** immune response, cell communication, cell growth and maintenance, protein metabolism, and regulation of nucleic acid metabolism.	[64]
G.S. Pessoa (2019)	19 BD; 19 SCZ; 13 CTR	Serum	Pooled	No	LC-MS/MS and LC/ICP-MS		MS	No/No	**BD vs. CT:****↑** (P01834; P0DOY2; J3QRN2; P01860; A0A0A0MTQ6; P01717; P01859) **↓** (P01857; P02787; P01620; S4R460)	Imbalance in the homeostasis of important micronutrients.	[65]
Y.H. Cheng (2018)	57 BD; 26 CTR	Serum; plasma: PBMCs	Individual	Yes	MALDI-TOF-MS	ELISA (P19882; Q95YL7) Flow cytometry (P38910)	MS	No/Yes	**BD vs. CT:****↑** (P19882; Q95YL7) **↓** (P38910)	Heat shock proteins (HSP) might be useful as a biomarker of BD and for distinguishing BD patients with abnormal HPA axis activity vs. normal HPA axis activity.	[66]
B. Petrov (2018)	12 BD; 11 MDD; 13 CTR	Serum	Pooled	No	LC-MS/MS	ELISA and WB (P02774)	MS	No/Yes	**BD:****↑** (P02774; P07357; P02745; P02747; P02746; P09871; P13671; P02776; P07996; P04275; P12259; P03952; P01008; P00747; P04004; P68366; Q9BQE3; Q9H4B7; P06396; P12814; Q13201; P08514; P05106; P37802; Q86UX7; P08185; P02760; P02753; P25311; Q9UGM5; P01042; Q96IY4; P22352; P30041; Q01518; P80108; P02749; P02655; P02647; P02656; P06727; P02649)	Inflammatory response	[67]
C. Knochel (2017)	25 BD; 29 SCZ; 93 CTR	Plasma	Individual	No	LC-MS/MS (MRM mode)	MRI	MS	No/No	**BD vs. CT:****↑** (P08697; P01008; P02647; P02652; P06727; P02654; P02655; P02656; P55056; P05090; P02649; Q13790; P00751; P01024; O75636; P05546; P14780; P36955; P02753) **BD vs. SCZ:** **↑** (P08697; P01008; P02647; P02652; P06727; P04114; P02654; P02656; P05090; Q13790; O14791; P00751; O75636; P05546; P04196; P36955; P02753); **↓** (P02655; P55056; P02747; P01024; P05160; P03952; P14780)	Altered APOC expression in BD and SCZ was linked to cognitive decline and underlying morphological changes in both disorders.	[33]
J.R. De Jesus (2017)	14 BD; 23 SCZ; 4 OD; 12 CTR (3 HCF; 9 HCNF)	Serum	Pooled	No	LC-MS/MS		2D DIGE	Yes/No	**BD vs. HCNF:****↑** (P02768; P02647); **↓** (P0C0L4; P01009; P02647; P02649) **BD vs. HCF:** **↑** (P02647); **↓** (P02786) **BD vs. OD:** **↓** (P02768; P0C0L4; P04004; P02656) **BD vs. SCZ:** **↓** (P0C0L4; P0C0L5; P02743)	An association between BD and altered immune and inflammatory functioning may be a probable mechanism that may explain the BD pathophysiology.	[32]
J.J. Ren (2017)	30 BD; 30 MDD; 30 CTR	Plasma	Pooled	Yes	LC-MS/MS		MS	Yes/No	**BD vs. CT:****↑** (Q0KKI6; Q86TT1; D6RD17; P20851; Q9UK54; Q9UL88; P04040; A0A0K2BMD8; P32119; P00915; P00441; B7Z2I6; P00738; Q6J1Z7; P01023; P30043; A0A068LKQ0; B3VL17; P01625; B3KRY3; Q9NP10; Q8N355; Q15430; Q6VFQ6; R4GN98; A0A0A0MSI0; P02763; P02647; A2KBC1; Q9NZD4; B7Z3I9; Q0ZCH9; F5H5I5; A0A0K0K1L1; Q13228); **↓** (B4E324; P80723; P31150; Q13103; H7C0V9; Q5T9B9; B2RAN2; U3KQE7; A0A0G2JS21; K7ESA0; A9X7H1; Q6UWP8; A0A075B6G4; A0A075B737; Q6ZRP7; J3KQ45; Q13201; P02775; Q8IUC0) **BD vs. MDD:** **↑** (P02763; Q9UBG0; P03973); **↓** (B4E1B2; B2RAN2; P02647; Q5T9B9; Q6UWP8; Q6ZRP7)	B2RAN2 and ENG with important roles in oxidative stress and the immune system may serve as candidate biomarkers for distinguishing MDD and BD.	[68]
Y.R. Song (2015)	45 BD (10 euth; 20 dep; 15 man); 20 CTR	Plasma	Pooled	No	MALDI-TOF/TOF MS	WB (P02647; O14791; P00915; P02743; P01023)	2-DE	Yes/No	**Eut. BD vs. CT:****↑** (Q6PEJ8; O14791; P43652; P36955; Q6U2M2; V9H0D6; Q96IY4; P02787; P02675); **↓** (P02647; P02774; P15169; Q96PD5; P19827; P02743; Q14624) **Dep. BD vs. CT:** **↑** (Q6PEJ8; O14791; P43652; P36955; Q6U2M2; P02774; V9H0D6; Q96IY4; P02787; P02675; P02743; P02679; P01024; P02790; P04264); **↓** (P02647; P15169; Q96PD5; P19827; Q14624; O43866; P04003; P00915) **Man. BD vs. CT:** **↑** (Q6PEJ8; O14791; P43652; P36955; Q6U2M2; P02774; V9H0D6; Q96IY4; P02787; P02675; P35527; P02747); **↓** (P02647; P15169; Q96PD5; P19827; P02743; Q14624; P01023; Q03591; P00736; P02671)	BD pathophysiology may be associated with early perturbations in lipid metabolism that are independent of mood state.	[49]
J. Chen (2015)	20 BD II; 30 MDD; 30 CTR	Plasma	Pooled	Yes	MALDI-TOF/TOF MS	ELISA (A8K2H7; P05156; P04003)	2-DE	No/No	**BD vs. MDD:****↑** (P02765; P02765; P04004; P04217; Q9BYX7; A2AJT9; isoform KNG1^#^; isoform HPX^#^); **↓** (Q03933; P27169; P06727; O75116; P01024; P02743; P63261; Q8WXH0; D3DP16; P05156; P01871; P02790; Q96PD5; P04003; isoforms KNG1^#^)	Immune regulation, including defense response, acute inflammatory response, response to wounding and inflammatory response.	[69]
L. Giusti (2014)	15 acute BD; 11 MDE; 15 CTR	PBMCs	Individual	No	LC-MS/MS	WB (P16219; Q14847; O43399; P31948)	2-DE	No/No	**BD vs. CT:****↑** (P02787; P18206; P02768; P31948; P10809; P02675; P08670; P07437; P01871; P00738; P16219; P14618; P0C0L4; P27482; Q14847; O43399; P15259; P60174; P02647; P02766); **↓** (P60709; Q13347; P11177; O00299; P63104) **BD vs. MDE:** **↑** (P31948; P02675; P60709; P00738; P16219; P14618; Q13347; P11177; P27482; Q14847; O00299; O43399; P15259; P63104; P02647); **↓** (P02787; P18206; P02768; P10809; P08670; P07437; P01871; P0C0L4; P60174; P02766)	Differential expression of cytoskeletal and stress response proteins in PBMCs.	[70]
J. Iavarone (2014)	17 BD; 32 SCZ; 31 CTR	Saliva	Individual	No	LC-MS/MS		MS	No/No	**BD vs. CT:****↑** (P59665; P59666; P12838; P80511; P01040; P04080; DEF2 *)	Dysregulation of the immune pathway of peripheral white blood cells	[52]
M. Herberth (2011)	Serum: 32 euth BD (I/II: 16/16); 32 CTR. PBMCs: 16 BD (I/II: 8/8); 15 CTR	Serum; plasma	Individual	No	LC-MS/MS	Immunoblot analysis (P55072; Q99798)	MS	No/Yes	**Serum****BD vs. CT:****↑** (O15467; P25942; P29965; P29279; P05305; P01133; Q0VHD7; P14174; P47992; P01229; P08263; P18065); **↓** (P02647; Q9Y258; P01876; P01871; P35225; P21583; P01375; P02656) **PBMCs** **BD vs. CT:** **↑** (O75083; Q00610; Q14008; Q14152; Q2M1P5; Q96Q89; P52179; Q9UKX3; Q9UKX2; P12883; Q71U36; Q99798; Q96KP4; Q59G92; P00338; P22314; P62937; P14625; P11142; P08238; P55072; Q14687; Q86V48; Q8IVG5; B7ZMG3); **↓** (P07355; O15061; P35580; P35749; Q9Y623; Q9Y4I1; Q14980; Q96PE2; O95347; Q99666; Q5T200)	Markers of euthymic BD patients pointing towards an increased inflammatory response and cell death in the immune system, along with increased activation of HPG axis hormones.	[50]
A. Sussulini (2011)	25 euth BD (15 BD + Li; 10 BD − Li); 15 CTR	Serum	Pooled	No	SELDI-TOF MS	Immunoturbidimetric (P02647)	2D DIGE	Yes/No	**BD + Li vs. BD − Li:****↑** (P02647); **↓** (P04004; P02766; P01009; P01857; P01009; P01008)		[63]
A. Sussulini (2010)	25 euth BD (15 BD + Li; 10 BD − Li); 15 CTR	Serum	Pooled	No	MALDI-TOF MS/MS and LA-ICP MS		2D-PAGE	Yes/No	P23142; P09871; P04004; P10909; P02743; Q96LC7; P02647; P02766; P0C0L4 (qualitative analysis)		[62]

BD I, bipolar disorder type I; BD II, bipolar disorder type II; euth, euthymic; dep, depressive; man, maniac; CT, controls; HCF, familiar healthy control; HCNF, non-familiar healthy control; BD + Li, BD patients treated with lithium; BD − Li, BD patients treated with other drugs; SCZ, schizophrenia; OD, other disorders; MDD, major depressive disorder; MDE, major depressive episode; WB, Western blot; MRI, magnetic resonance imaging.^#^ Proteins represented by entry name (isoforms); * Protein represented by code name (no information about protein ID was found; we could not find the corresponding accession number/identifier through the UniProt database).

## 4. Main Studies Performed

### 4.1. Bipolar Disorder vs. Control

In all the selected studies (*n* = 14), a comparison between BD and a control group was performed, expressing the main objective of establishing a proteomic profile of the disorder and potentially identifying specific proteins that could help in the diagnosis.

Overall, 258 proteins were identified as differentially expressed between BD patients and clinical controls in blood-related samples (plasma, serum, and PBMCs). The summary of the number of proteins identified as altered is shown in Figure 2 (see also Appendix A). In total, 105 proteins were identified as altered in plasma, 115 proteins in serum, and 62 proteins in PBMCs. 

Some proteins identified as altered in BD were coincident in the different blood-related matrices. In fact, three proteins were identified in all the three matrices, namely apolipoprotein A-I, complement C4-A, and serotransferrin (accession numbers P02647, P0C0L4, and P02787, respectively); 12 proteins were identified in plasma and serum: antithrombin-III, apolipoprotein A-IV, apolipoprotein C-II, apolipoprotein C-III, apolipoprotein D, apolipoprotein E, carboxypeptidase B2, complement C1q subcomponent subunit C, multimerin-1, plasma kallikrein, retinol-binding protein 4, and vitamin D-binding protein (accession numbers: P01008, P06727, P02655, P02656, P05090, P02649, Q96IY4, P02747, Q13201, P03952, P02753, and P02774, respectively); two proteins were coincident between plasma and PBMCs: fibrinogen beta chain, and haptoglobin (accession numbers: P02675, and P00738, respectively); and four proteins were identified in both serum and PBMCs actin cytoplasmic 1 or 2, Ig mu chain C region, serum albumin, structural maintenance of chromosomes protein 2 (accession numbers: P60709, P01871, P02768, and O95347, respectively).

In the study using saliva samples, eight proteins were identified as altered [52]. However, the identified proteins were not the same when comparing saliva and blood-related samples.

### 4.2. Bipolar Disorder vs. Schizophrenia

In total, five studies had a comparison between BD and SCZ, with higher prevalence after 2017 (*n* = 4) [32,33,64,65], against only one study before that date [52]. The BD vs. SCZ studies prevalence in the last years reflects an increasing interest in identifying differentially expressed proteins between these two major mental disorders towards the definition of disorder-specific biomarkers to help with the diagnostic specificity.

The four BD vs. SCZ studies published after 2017 were performed in blood-related samples, namely plasma and serum. Two studies were performed in serum samples, enabling the identification of 48 altered proteins [32,64]. One study focused on plasma samples, with the identification of 25 altered proteins [33], and the fourth study performed the comparison of the ionomic profile of BD and SCZ disorders against a control group towards establishing relationships between metals and proteins [65]. 

Overall, 70 differentially expressed proteins were identified in the comparison between BD and SCZ patients. By assessing these proteins, it was observed that three proteins were coincident in plasma and serum samples: apolipoprotein D, apolipoprotein E, and retinol-binding protein 4 (accession numbers: P05090, P02649, and P02753, respectively). Interestingly, these three proteins were also highlighted in BD vs. control studies, being altered both in plasma and serum samples.

In the study using saliva samples, eight proteins (α-defensin 1–4, S100A12, cystatin A and S-derivatives of cystatin B, cystatin B S-glutathionyl, and cystatin B S-cysteinyl) were identified as altered between BD and SCZ against control; however, no statistically significant differences were observed between the SCZ and the BD groups [52].

### 4.3. Bipolar Disorder vs. Other Disorders

Five studies compared the proteome of BD patients and other disorders (OD): three studies of major depressive disorder (MDD) [67,68,69], one study of major depressive episode (MDE) [70], and one study described as other mental disorders [32]. All these studies were published after 2014, being consistent with the trend observed in BD vs. SCZ studies (interest in the definition of disorder-specific biomarkers towards helping the diagnostic specificity).

No tendency to analyze a specific blood matrix was observed. Plasma samples were used in two studies [68,69], two studies analyzed serum [32,67], and one study was performed on PBMCs samples [70].

Overall, 54 proteins (see Appendix A) were identified as altered in the comparison between BD and other mental disorders. Four proteins were identified in serum [32], 31 proteins were identified in plasma [68,69], and 25 proteins were identified in PBMCs [70]. Some proteins were identified as altered in more than one blood-related matrix: one protein in plasma and serum: vitronectin (accession number P04004), two proteins in plasma and PBMCs: apolipoprotein A-I, and Ig mu chain C region (accession numbers P02647 and P01871, respectively) and two proteins in serum and PBMCs: complement C4-A, and serum albumin (accession numbers P0C0L4 and P02768, respectively). The pairs of proteins apolipoprotein A-I and complement C4-A and Ig mu chain C region and serum albumin had already been highlighted in the comparison of BD vs. control, with the first pair of proteins identified in the three blood-related samples (plasma, serum, and PBMCs) and the second pair in serum and PBMCs.

In Rhee and colleagues [72], a study published in 2020, the profile of BD and MDD subjects was compared by analysis of serum by LC-MS/MS, allowing the identification of fourteen differentially-expressed proteins between the disorders. The proteins ras-related protein Rab-7a, rho-associated protein kinase 2, Ig heavy chain V–III region WEA, monocyte differentiation antigen CD14, Ig heavy chain V–I region V35, Ig heavy variable 3–64, and Ig heavy chain V–III region JON were significantly overexpressed in subjects with MDD; whereas exportin-7, angiogenin, Ig lambda variable 3–25, serum amyloid A protein, coagulation factor XIII B chain, and selenoprotein P were significantly overexpressed in subjects with BD. Although we acknowledge the relevance of these results, the study did not fulfill all eligibility criteria because it lacked a control group without mental disorders (one of the eligibility criteria for this systematic review) and, therefore, was not included in this review. 

### 4.4. Bipolar Disorder Patients Treated with Li-Drugs vs. Treated with Other Drugs

Two studies assessing BD patients’ treated with lithium and treated with other drugs were selected [62,63]. The same research group performed these two studies in 2010/2011. In Sussulini and colleagues [62], published in 2010, a metallomics study was performed to identify the main differences in metal-containing proteins to discover potential markers for BD and its treatment with lithium. The same author published in 2011 [63] a study where the proteome profile of BD patients and its treatment with lithium were assessed in order to identify characteristic signatures and, consequently, potential marker candidates.

In summary, considering the studies on BD, namely BD vs. control, BD vs. SCZ, and BD vs. OD, a total of 271 unique proteins were identified as altered in BD patients (Figure 3 and Appendix A). Discriminating that number, (i) 156 proteins were only identified in BD vs. control studies; (ii) one protein in BD vs. SCZ, (iii) 12 proteins in BD vs. OD; (iv) 60 proteins were identified in both BD vs. control and BD vs. SCZ; (v) 33 proteins were identified in both BD vs. control and BD vs. OD; and (vi) nine proteins were identified in all comparisons (BD against control, SCZ and OD), being these proteins: actin cytoplasmic 1, actin cytoplasmic 2, apolipoprotein A-I, apolipoprotein A-IV, apolipoprotein C-III, antithrombin-III, complement component C3, complement C4-A, serum amyloid P-component (accession numbers P60709, P63261, P02647, P06727, P02656, P01008, P01024, P0C0L4, and P02743, respectively).

### 4.5. Bias Analysis

The results of the evaluation of the quality of the proteomic studies included in the systematic review are displayed in Appendix A. The least fulfilled QUADOMICS quality criteria were the items 4 (6 studies), 11 (6 studies), and 16 (2 studies). The majority of the studies did not report enough data to assess items 6 and 12. Futhermore, one study [69] did not show sufficient quality information to be accurately assessed.

### 4.6. Meta-Analysis

Eight out of the 14 studies [32,33,50,52,63,65,68,70] included in the systematic review reported data in a format amenable with meta-analysis, i.e., they provided data about the differently expressed proteins between groups in the form of effect size (average or fold change) and error deviation (standard deviation or *p*-value).

It was only possible to compute the effect size (expressed in log2 fold change) in at least two independent studies, comparing bipolar patients with healthy controls for three proteins: apolipoprotein A-I, apolipoprotein C-III, and apolipoprotein E (accession numbers P02647, P02656, and P02649, respectively). A meta-analysis included these proteins to assess their overall expression change (forest plot in Figure 4).

There was no overall statistically significant difference in the expression of these proteins between bipolar patients and healthy controls, although for apolipoprotein A-I three out of four studies showed an increase in BD patients. It should be mentioned that, for the three proteins included in the meta-analysis, the degree of protein quantification heterogeneity was high between these studies, with no consistent results in the protein fold change tendency.

## 5. Discussion

In this work, a comprehensive systematic review was performed along with a meta-analysis to evaluate mass spectrometry-based proteomics applied to human peripheral fluids to assess biomarkers of bipolar disorders and the identification of relevant networks of biological pathways.

The major studied topic was the assessment of protein expression differences between BD patients and healthy controls (CTR). A total of 266 proteins differently abundant were identified in peripheral fluids, including serum (115 proteins), plasma (105 proteins), PBMCs (62 proteins), and saliva (8 proteins).

Apolipoproteins (APOs) were one of the groups of proteins most found in BD vs. control studies as differentially expressed. In fact, nine studies reported the dysregulation of apolipoproteins [32,33,49,50,63,64,67,68,70]. Apolipoproteins are amphipathic molecules capable of interacting with both the lipids of the lipoprotein core and the aqueous environment of the plasma, having a well-established role in the transport and metabolism of lipids; and in inflammatory and immune response regulation [73,74,75]. Several studies have indicated apolipoproteins as potential candidates for psychiatric biomarkers, with extensive evidence that the levels of cholesterol and APOs may be disturbed in psychiatric disorders [48,74,76]. Accordingly, in the selected studies, APO alterations (in plasma, serum, peripheral blood mononuclear cells, and peripheral lymphocytes) were associated with inflammatory response [32,50,70] and lipid metabolism [63]. APO alterations were also related to cognitive decline and underlying morphological changes [33] and fat and vitamin digestion and absorption.

Apolipoprotein A1, APOA1, a major protein associated with lipid metabolism, was the apolipoprotein found as significantly altered in BD patients in more studies (total of seven), three in plasma [33,49,68], three in serum [32,50,63] and one in PBMCs [70]. APOA1 is a constituent of the HDL fraction, and regulates plasma levels of free fatty acids, and has an important role in HDL and triglyceride-rich lipoprotein metabolism and in the reverse cholesterol transport pathway [63]. However, APOA1 showed a heterogeneous behavior in the selected studies, expressed in an increasing behavior in five studies [32,33,63,68,70] and a decreasing one in two studies [49,50]. APOE, a protein with a critical function in lipoprotein-mediated lipid transport, was identified as differentially expressed in four studies, three in serum and one in plasma, being increased in three studies [36,65,68] and decreased in one study [32]. APOC3, an apolipoprotein capable of inhibiting lipoprotein lipase and hepatic lipase with genes closely linked with APOA1 and APOA4 in the human genome, was also identified in three studies as significantly altered, being increased in two studies [33,67] and decreased in one study [50]. Four APOs were identified in two independent studies; APOA4 and APOC2 [33,67] showed similar behavior in the two studies, expressing an increase in BD patients, whereas APOD and APOL1 showed a heterogeneous behavior. APOD was identified as up-regulated in the study of Smirnova et al. [64], which was found in BD serum and absent in the controls and SCZ and down-regulated in the study of Knochel et al. [33]. On the other hand, Knochel et al. found APOL1 up-regulated [33], while Song et al. reported a down-regulation [49].

Only APOs (APOA1, APOC3, and APOE) fulfilled the parameters and were assessed in the meta-analysis (Figure 4). In accordance, no clear alteration trend was observed for these proteins, with different behavior being identified in the selected studies.

A set of four proteins was also identified in three selected studies as differentially expressed in BD patients, namely serotransferrin, TRFE [50,66,71], complement C1q subcomponent subunit C, C1QC [33,49,67], complement C4-A, CO4A [32,49,70], and retinol-binding protein 4, RET4 [33,64,67].

Transferrin (TRFE) is the main extracellular iron transport protein in blood that, when bound with iron, interacts with exofacial transferrin receptors allowing internalization of that complex of iron and proteins into cells [77], being related to perturbations of iron uptake in subjects with a major depressive disorder associated with immune response and pro-inflammatory changes [70,78]. TRFE was found in two studies to be significantly increased in BD patients [49,70] and decreased in one study [65], being associated with an inflammatory response in one of the studies [70].

CO4A and C1Q are proteins involved in the complement system, which is a part of the innate immune system participating in the regulation of inflammation [79,80].

CO4A is an essential component of the effector arm of the humoral immune response, playing a central role in activating the classical pathway of the complement system and mediating inflammatory processes. CO4A was found as increased in two studies [49,70]. In Song et al., higher levels of complement factors, such as plasma C3 or C4 concentrations, as well as acute-phase proteins and a moderate increase of pro-inflammatory cytokines, were identified in BD patients suggesting that chronic, mild inflammatory processes in the peripheral and central nervous system (neuroinflammation) are involved in BD pathophysiology.

C1Q is a complex glycoprotein that mediates a variety of immunoregulatory functions considered important in preventing autoimmunities, such as enhancing phagocytosis, regulating cytokine production by antigen-presenting cells and subsequent alteration in T-lymphocyte maturation [81]. C1Q was found to increase in two studies [49,67], whereas in another study, it was found to decrease, although slightly (fold change = 0.99) and not statistically significant [33]. In Song et al. [49], a comparative proteomic study was performed to identify differentially expressed proteins in various BD mood states (depressed BD, manic BD, and euthymic BD) relative to healthy controls, an increased behavior of C1Q was only found in the maniac state, suggesting that C1Q may be involved in the pathophysiology of maniac episodes.

RET4 is mainly expressed in the liver with a primary function of transporting retinol (vitamin A) from the liver to peripheral tissues, with retinol being essential for the brain to facilitate learning, memory, and cognition [82]. The RET4-retinol complex interacts with transthyretin and prealbumin, increasing the serum half-life of RET4 [83]. It is known that retinoid signaling plays a significant role in immune cell function; thus, it is suggested that factors affecting this system could have important implications for schizophrenia and other psychiatric disorders-associated inflammatory stress [83]. RET4 was found as increased in three studies [33,64,67]. The information was consistent; however, only one of those studies could compute the effect size [33], highlighting the lack of complete statistical information in some selected studies (six out of 14) in a format amenable to a meta-analysis.

Following the same trend observed in APOs results, the trends for these proteins were also heterogeneous, with the different studies reporting a distinct protein behavior. This is an important observation, highlighting the urgent need to standardize proteomics strategies to obtain meaningful and consistent information about BD. 

The immune system and inflammatory response were the most identified biological processes being altered in BD patients [32,49,50,52,64,66,67,68,69,70]. These findings are consistent with current knowledge about BD, with the immune system and inflammatory response being strongly associated with the BD pathophysiology [21,84,85]. Several mechanisms have been identified to explain the relationship between BD and immune dysfunction, including cytokine-induced monoamine changes, increased oxidative stress, pathological microglial over-activation, hypothalamic–pituitary–adrenal (HPA) axis over-activation, alterations of the microbiome–gut–brain axis and sleep-related immune changes [21]. In accordance, the regulation of the biological processes known as transport, morphological changes, cognitive impairment, oxidative stress lipid metabolism and activation of HPG (hypothalamic–pituitary–gonadal) axis hormones have been identified as altered in BD patients [86,87,88].

In order to validate these findings and integrate the biological meaning of the results from all the studies presented, a gene ontology (GO) analysis was performed where all proteins that were found to be altered in any of these studies were used (Figure 5 and Appendix A).

From this ontological analysis, it is possible to observe that many proteins were found to be altered throughout the studies (264 proteins, two of the proteins identified as altered in the saliva study [52] were not included in the gene ontology analysis as no accession number was available), but only 29 proteins (roughly 14%) were found to be altered in more than one study. From these 29 proteins, only 12 proteins are reported as having similar regulation trends in BD, in this case, up-regulation for all 12. The other 17 proteins are reported with contradictory regulations between the studies (Appendix A).

Several pathways were highlighted by performing this functional analysis, as shown in Figure 5 and Appendix A. The result of this analysis shows that metabolic pathways such as β-alanine or pyruvate metabolism (along with glycolysis or gluconeogenesis and the TCA cycle) are among the pathways with higher enrichment *p*-value and pathway impact. This metabolic dysregulation of BD patients has been consistently reported in the literature, although with sometimes conflicting results, along with oxidative stress and the dysregulation of antioxidant systems [89,90] also highlighted in this list with terms as glutathione or riboflavin metabolism. 

Another pathway that is enriched by the proteins considered altered within all the studies is the complement and coagulation cascades, which present the lowest *p*-value (and FDR corrected *p*-value) for the enrichment of the pathway. Some of the proteins considered as altered in this pathway, with special emphasis on the complement cascade, have already been discussed above and are among the proteins that are consistently found to be altered in several studies. Many of these proteins are also related to immune response, and these pathways are also highlighted by this gene ontology analysis with terms related to bacterial infection or auto-immune disorders. Interestingly, proteins implicated in the complement and coagulation cascade have been highlighted in recently proposed proteomic prediction models of transition to clinical psychosis [91].

As it has been discussed above, among the proteins most widely reported as altered in BD are apolipoproteins being the cholesterol metabolism one of the pathways presented with a lower enrichment *p*-value. Several apolipoproteins are reported altered in more than one study, and although some are reported with inconsistent results (see meta-analysis above), others are reported as up-regulated in all studies. In particular, lipid metabolism and cholesterol metabolism have long since been associated with brain disorders, including BD, and some supplementation to the medication has already been reported as beneficial for the treatment of the disorder [92].

Another pathway highlighted by this analysis, with high impact and low enrichment *p*-value, is focal adhesion (Appendix A), where several key proteins were found to be altered throughout the studies, but only actin was reported in common in more than one study (with inconsistent results). This pathway has already been related to BD in other studies [93], and here in this analysis, it has a high impact. Thus, it may be interesting to investigate and validate the potential of these proteins in BD diagnosis or physiology.

This ontological analysis focused primarily on proteins that were found to be altered between BD and control individuals, as this is the most widely reported comparison. Nonetheless, other comparisons are performed throughout the studies with as much or even more interest in investigating BD biomarkers and pathophysiology.

The recent interest in the comparisons of BD vs. SCZ and BD vs. other disorders studies reflects the increasing concern on the definition of disorder-specific biomarkers and the understanding of the associated altered biological pathways to improve diagnostic specificity. 

Five studies had a comparison between BD and SCZ [32,33,52,64,65], showing a strong heterogeneity of the type of biological sample analyzed, serum [32,64,65], plasma [33], and saliva [52]. Following the same trend observed in BD vs. CTR studies, the immune system and inflammatory response were the most highlighted biological pathways [32,52,64].

In the study performed by Smirnova et al. [64], the definition of the proteome profiles of different groups revealed 27 proteins specific for schizophrenia (not present in BD) and 18 for BD. The specific proteins of schizophrenia were mostly associated with immune response, cell communication, cell growth and maintenance, protein metabolism and regulation of nucleic acid metabolism, whereas the protein set in BD was mostly associated with immune response, regulating transport processes across the cell membrane and cell communication, development of neurons and oligodendrocytes and cell growth.

In de Jesus et al. [32], three unique proteins (CO4A, CO4-B, and SAMP) were identified as differentially abundant between BD vs. SCZ, being associated with the inflammatory response. All these proteins were found with lower abundance in the serum of BD patients when compared with serum from SCZ patients.

In the study performed by Knochel et al. [33], the plasma protein expression in BD and SCZ patients was associated with cognitive deficits and their underlying brain structures. BD patients were found to have higher concentrations of the proteins A2AP, ANT3, ApoB, ApoD, and ApoF (all with *p* < 0.05). The results suggested that detecting molecular patterns in association with cognitive performance and its underlying brain morphology was important to better understand the pathological mechanisms of BD and SCZ and, consequently, to support the diagnosis and treatment of both disorders.

The study using whole saliva samples [52] confirmed a schizophrenia-associated dysregulation of the immune pathway of peripheral white blood cells and suggested that the dysregulation in the BD group could involve the activation of more specific cell types than that of the SCZ group. The significant increase in salivary concentrations of the eight proteins reported in SCZ and BD subjects confirms that these patients display strong innate immune system activation. Moreover, the very significant and higher correlations found between the levels of these proteins in BD subjects with respect to the SCZ group strongly suggest that the immune system activation in BD subjects could be linked to more specific cell types than in SCZ subjects.

In Pessoa et al. [65], a metalloproteomics study was performed, allowing the identification of the proteins IGHG1 (both BD and SCZ), IGKV2D-28 (only in BD), and Ig lambda chain V-IV region Hil and ApoH (only in SCZ) as altered in BD and SCZ comparing to a healthy group and the identification of different concentrations of Li, Mg, Mn, and Zn in BD patients and high levels of Cu for SCZ patients, indicating an imbalance in the homeostasis of important micronutrients.

The comparison of BD vs. other disorders (OD) was assessed in five studies, reflecting comparisons with major depressive disorder (MDD) [67,68,69], major depressive episode (MDE) [70], and one study described as other mental disorders [32]. Consistently with the previously mentioned comparisons, a strong heterogeneity of biological samples was also used, with the analysis of serum [32,67], plasma [68,69], and PBMCs [70].

In Ren et al. [68], the proteomics profiles of plasma samples were compared to differentiate between BD and MDD in the first depressive episode because of the potential treatment implications. In this study, nine proteins were identified as significantly altered between MDD and BP, most of them related to the immune system. Particularly relevant, given the important role of B2RAN2 (highly similar to vanin-1) and endoglin in oxidative stress and the immune system, they may play roles in depression, being suggested as potential candidate biomarkers to distinguish between MDD and BP.

The association between vitamin D and inflammatory markers in adolescents with BD and MDD was studied by Petrov et al. [67], allowing the identification of higher levels of D-binding protein (DBP) in BD patients and its suggestion as a marker candidate for BD.

BD is closely related to inflammatory processes, which were thoroughly investigated for their role in BD and MDD in the context of major inflammatory markers such as IL-6, MCP1, and TNFα. Accordingly, with the relevance of inflammatory response identified by de Jesus et al. [32], the distinction between BD and OD was assessed, allowing to identify four proteins (VTN, ALB, CO4A, and APOC3) significantly altered and also associated with the inflammatory response.

## 6. Strengths and Limitations

Mass spectrometry (MS)-based proteomics strategies have a (semi)-quantitative character, generating different levels of information about the individual proteome.

In recent years, advances in MS technologies combined with more sensitive and selective sample preparation techniques and with increased computer data processing capability and new data processing tools allow dealing with the large amount of information generated by MS approaches. In addition, making use of computational advances, the development of novel data analysis strategies (e.g., statistical and machine learning approaches) also increased the ability to extract meaningful information from MS data, facilitating the mechanistic understanding of the biological processes associated with the disorder.

From a clinical point of view, the study of human peripheral fluids represents an attractive approach. Peripheral fluids contain disease-associated proteins secreted or leaked from pathological tissues across the body, easily obtained through non-invasive procedures that allow a large sample volume collection [94].

The main limitations that could be pinpointed in these selected studies are related to the huge heterogeneity observed in the cohorts’ characteristics. In fact, the different studies reported here show significant differences related to diagnostic criteria used, medical and psychiatric comorbidities reported, psychopharmacological treatment and its duration, as well as the sociodemographic variables such as age, illness duration, gender balance, diet, and use of tobacco, alcohol or other psychoactive substances, among others.

The diagnostic criteria from the two main classification systems applied in the selected studies, the DSM-IV and the ICD-10 criteria, have a major difference between them. For DSM-IV criteria, a distinction between type I and type II is made, defining BD I with at least one episode of full-blown mania or mixed episode; whereas ICD-10 criteria do not discriminate between BD I and BD II, with BD diagnosis consisting of two discrete mood episodes with at least one of which being maniac [95].

The majority of the studies excluded BD patients with inflammatory, autoimmune, hepatic, endocrinological, and metabolic diseases (e.g., cancer, AIDS, diabetes, heart disease) and other psychiatric disorders; however, there is a huge discrepancy in the details given for the exclusion criteria. While in some studies, there is a clear definition of the diseases that lead to a patient’s exclusion, in other studies, there is only a subjective description. In addition, only in a few studies was pregnancy mentioned as an exclusion condition.

Similarly, BD patients’ exclusion based on sociodemographic variables, such as the use of tobacco, alcohol, or other psychoactive substances and diet, and blood collection conditions (e.g., hemolysis, piarhemia, or defibrinated blood) are also reported in some studies. However, there is no standardization of the conditions applied, leading to high heterogeneity between the different studies. A clearer and tighter definition of the exclusion criteria and the sociodemographic conditions is essential, seeking to minimize the confounding factors that may hinder the identification of potential proteomic changes specific to BD and the use of the data in further meta-analysis.

The psychopharmacological treatment and its duration characteristics reported in the studies are also highly heterogeneous, adding confounding variables to the study. The characteristics found in the 14 selected studies varied from drug-naïve/minimally medicated (not receiving psychopharmacological treatment for at least 1–6 months prior to sample collection) to medicated. Studies with medicated BD patients are the most common, although drug naïve/minimally medicated BD groups have been increasingly used in the last five years, reflecting the concern on minimizing potential confounding factors. Here a clear definition of the research objectives is essential. The use of a minimal medicated BD group is important for a more comprehensive understanding of the pathophysiology of BD and the identification of potential biomarkers of the disorder. However, studies related to the effects of psychiatric drugs to treat BD and the definition of differences in the proteomic profile of patients with BD with those drugs are also essential.

Comparing the average age of BD with the healthy control groups, it was similar between each group in each study. The care of using similar average age between groups and sharing age characteristics of the groups in the more recent studies reveals a research concern in the homogeneity and detailed description, at least in the age parameter.

BD cohorts’ average illness duration varied between 2 and 13 years, with six studies having average values above 4.5 years. Similar to the age parameter, in illness duration there is a recent awareness of the importance of sharing detailed information, with five of the six studies that did not share illness duration information prior to 2016. 

For gender comparison, no clear trend was identified in the selected studies, although more studies had more females for the BD and the control groups, six out of 14 for both groups. Also, a significant number of studies did not share gender information (four studies for the BD group and six for the control group). Using gender-balanced groups is important to minimize confounding factors associated with gender. A predominance of maniac component both at onset and during their lifetime occurs commonly in men, with earlier onset of their illness, and alcohol and other drug abuse comorbidities [96,97,98]. On the other hand, in women, there is a tendency for a predominance of depression at onset and through their lifetime; usually, they experience more mixed manic episodes. When compared to men, BD onset in women often occurs at an older age, with greater comorbidities of physical pathology (particularly thyroid disease) and eating and anxiety disorders, and higher adherence to medication [97,98]. Women can also experience two significant periods that can be critical periods for BD: pregnancy and postpartum [98]. Since the different gender present distinct BD characteristics, gender studies are also important to better understand the pathophysiology of BD, although it is essential that BD and control groups have a balanced number of samples.

The strength of studying drug naïve/minimally medicated BD patients with no other medical and psychiatric comorbidities and clearly defined sociodemographic variables is that the potential confounding influence of medication and other diseases on the patient’s overall proteome is eliminated/minimized and, therefore, enabling a more comprehensive understanding of which differentially expressed proteins are related with the disorder. Accordingly, the definition of standardized sociodemographic, clinical and cognitive variables across the studied groups would lead to more objective and specific studies allowing a more comprehensive understanding of BD pathophysiology and, consequently, increasing the possibility of identifying specific biomarkers of BD. The study of medicated BD groups with the same standardized conditions will also be important to evaluate and define more accurate psychopharmacological treatment specific to each individual or individuals with specific characteristics.

## 7. Directions for Future Research

The recent advances in mass spectrometry proteomics approaches applied to human peripheral fluids allow the establishment of a robust platform for proteome profiling of clinical samples with an unprecedented depth. In fact, the MS ability to generate different levels of information about the individual proteome may lead to the comprehensive characterization of the biological network of pathways involved in BD, seeking the identification of reliable biomarkers of the disorder to improve prediction and diagnosis towards the ultimate goal of improving patient care and outcome. 

However, for more specific clinical proteomics studies, a standardization of the studies’ characteristics is required, seeking to minimize the confounding factors. In fact, the clear definition of the study’s objectives and standardization of sociodemographic, clinical, and cognitive variables across the studied groups would make them more objective and specific, allowing a more comprehensive understanding of BD pathophysiology and increasing the possibility of identifying specific biomarkers of BD. This will minimize the confounding factors, leading to improvements in the statistical power and, consequently, to the efficiency of translating biomarker candidates and drug targets to the clinical application associated with the disorder.

The use of proteomics pipelines combining (i) standardized study conditions; (ii) high-throughput sample preparation techniques; (iii) high computational power for data processing and analysis will lead to a rapid expansion of clinical cohort sizes and consequently to more robust studies. An extra effort should be made to provide data in an open format so the community can re-analyze and perform larger studies based on data analysis from multiple centers. After full implementation of those proteomics pipelines, their application in extended clinical cohorts will allow taking into account the different variables (such as gender, comorbidities, illness duration, and treatment), leading to a more comprehensive understanding of BD pathophysiology and, consequently, increasing the possibility of identifying specific biomarkers of BD, seeking to improve prediction and diagnosis towards the ultimate goal of improving patient care and outcome.

## 8. Conclusions

Our results highlight the potential of mass spectrometry clinical proteomic studies in the future search for biomarkers for BD, with the main biological pathways shown as enriched, including complement and coagulation cascades, lipid and cholesterol metabolism, and focal adhesion, among others. Our review also points to various factors that can contribute to the heterogeneity of the findings, including differences in sample size and characteristics, heterogeneity in the definition of the clinical phenotype, peripheral fluid sample preparation, analytical methods, and data analysis pipeline. The paucity of studies that employed a validation cohort and the lack of standards in statistical results reporting are also important pitfalls of previous studies. Further studies with broad samples and validation cohorts, longitudinal designs with patients in early phases of BD, and integrating proteomics results with other omics data (phenomics, genomics, metabolomics, connectomics) could provide additional information about differentially expressed proteins in selected biological pathways. The comprehensive information produced could harbor proteomic biomarkers of BD, contributing to individualized prognosis and stratification strategies, besides aiding in the differential diagnosis.

## Figures and Tables

**Figure 1 ijms-23-05460-f001:**
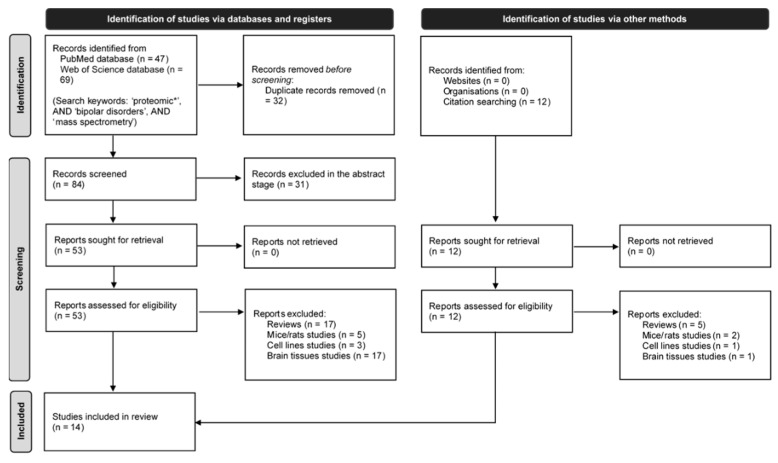
Flow diagram of the selection process of the studies included in the systematic review, following the directives of PRISMA 2020 [61].

**Figure 2 ijms-23-05460-f002:**
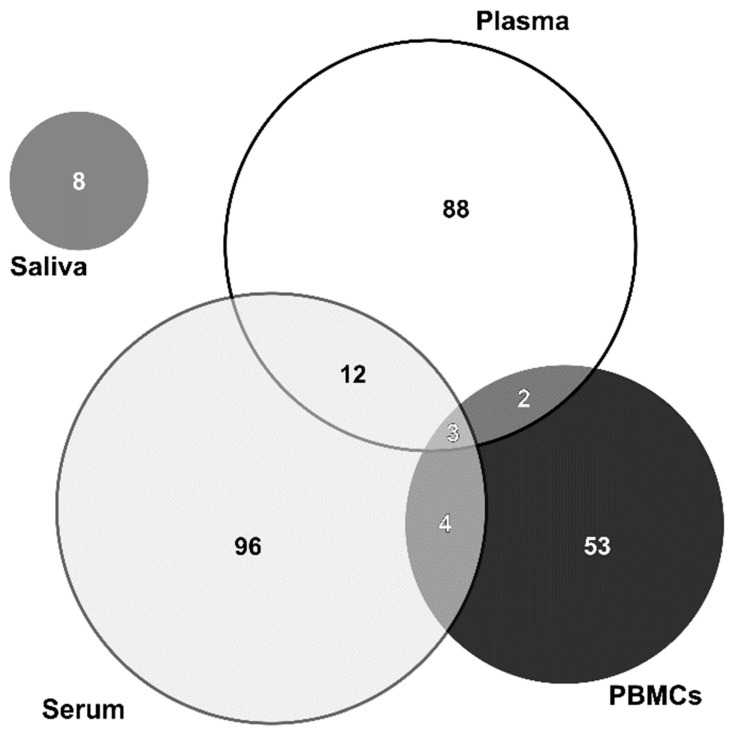
Venn diagram of proteins identified as altered in blood samples (plasma, serum, and PBMCs) and the whole saliva in the selected studies of bipolar disorders (BD) vs. control. The proteins identified as altered in the (i) plasma vs. serum vs. PBMCs (3 proteins), (ii) plasma vs. serum (12 proteins), (iii) plasma vs. PBMCs (2 proteins), and (iv) serum vs. PBMCs (4 proteins). The proteins found as altered in the saliva study were not coincidental with the proteins found in the blood studies (see text for details).

**Figure 3 ijms-23-05460-f003:**
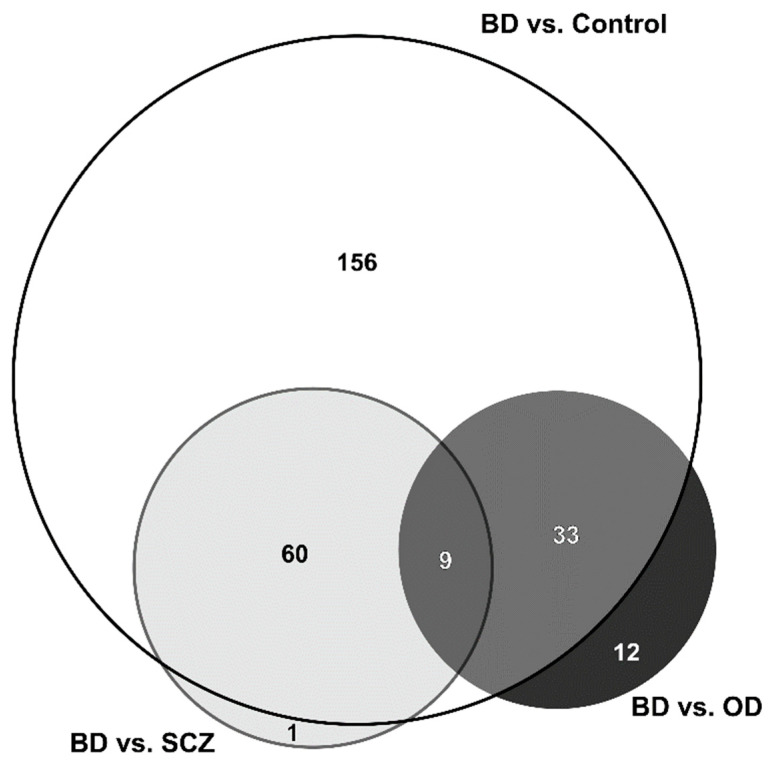
Venn diagram of proteins identified as altered in BD in the selected studies (BD vs. control, BD vs. SCZ, and BD vs. OD; see text for details).

**Figure 4 ijms-23-05460-f004:**
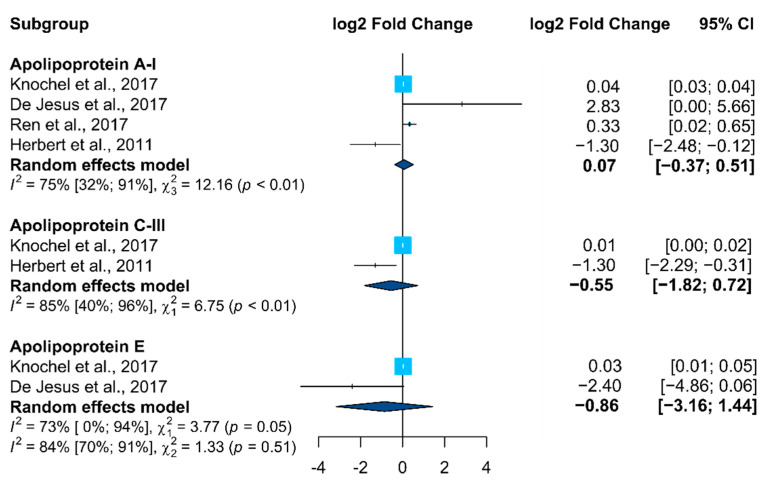
Forest plot from the meta-analysis of proteins identified as altered in BD vs. control studies in at least two studies (95% CI, confidence intervals). Squares (whiskers represent 95% CI) indicate the effect sizes of the individual studies. The size of the squares reflects the sample size of each individual study. Diamonds represent summary statistics.

**Figure 5 ijms-23-05460-f005:**
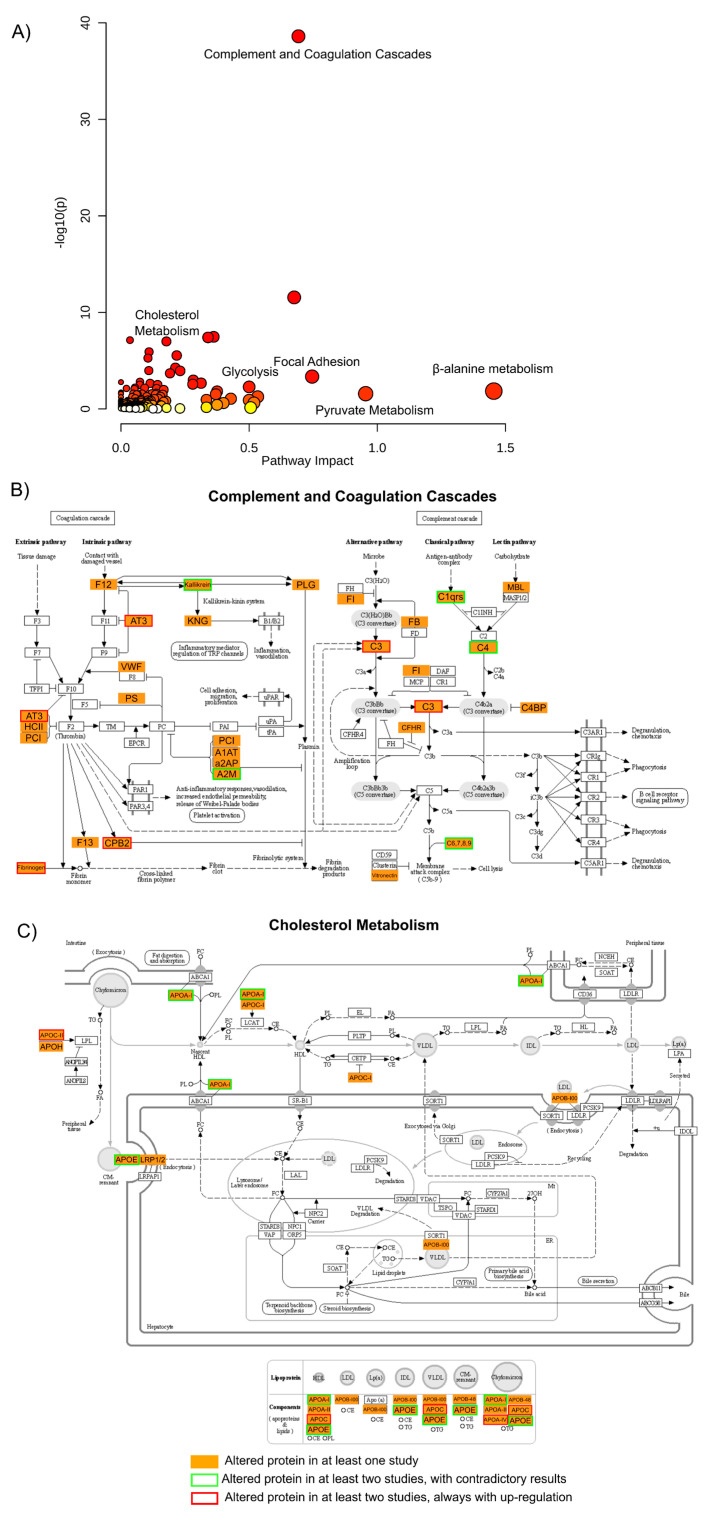
Gene ontology analysis of all altered proteins. (**A**) A gene ontology approach was used to assess pathway impact and enrichment (here presented by the *p*-value and color scheme) of the 264 proteins described as altered in BD vs. CTR in at least one study (Appendix A), represented here as a scatter plot [59]. From the pathways shown as enriched by this list of proteins, two were selected and their visual representation obtained through KEGG Mapper Color tool [60]: (**B**) complement and coagulation cascades, and (**C**) cholesterol metabolism. In these last two panels, the proteins found in any of the studies are shown in orange, and proteins found to be altered in at least two studies are highlighted in red when the protein is always found to be up-regulated in BD cases or highlighted in green when the results from the two or more studies are contradictory.

## Data Availability

Not applicable.

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
