# Peer review of "Systematic Review and Meta-Analysis on MS-Based Proteomics Applied to Human Peripheral Fluids to Assess Potential Biomarkers of Bipolar Disorder"

_ijms, 2022, doi:10.3390/ijms23105460_

Round 1

Reviewer 1 Report

1 April 2022

Regarding the review of manuscript “Systematic review and meta-analysis on MS-based proteomics applied to human peripheral fluids to assess potential biomarkers of bipolar disorder” by Rodrigues JE et al., submitted to International Journal of Molecular Sciences

Manuscript ID: ijms-1667369

Dear Authors,

Rodrigues and colleagues investigated in the present review entitled ‘Systematic review and meta-analysis on MS-based proteomics applied to human peripheral fluids to assess potential biomarkers of bipolar disorder’, the current status of knowledge of the efficacy of mass spectrometry (MS) proteomics applied to human peripheral fluids to assess bipolar disorder (BD) biomarkers and identify relevant networks of biological pathways. For this purpose, the authors identified fourteen articles, that allowed the identification of 266 differently expressed proteins, complement and coagulation cascades, lipid and cholesterol metabolism, and focal adhesion as the main enriched biological pathways. Although the proven ability of MS proteomics to characterize BD, there were several confounding factors contributing to the heterogeneity of the findings. The authors concluded by stating that in the future, it would be necessary to integrate omics with bioinformatics tools to provide a comprehensive understanding of proteome alterations, biomarkers of BD, and to contribute to individualized prognosis and stratification strategies.

The main strength of this original review article is that it addresses an interesting and timely question, investigating the potential of mass spectrometry clinical proteomic studies in the future search of biomarkers for BD. In general, I think the idea of this article is really interesting and the authors’ fascinating observations on this timely topic may be of interest to the readers of the International Journal of Molecular Sciences. However, some comments, as well as some crucial evidence that should be included to support the author’s argumentation, needed to be addressed to improve the quality of the manuscript, its adequacy, and its readability before its publication in the present form, in particular reshaping parts of the Introduction and Discussion sections by adding more evidence and theoretical constructs.

Please consider the following comments:

  1. Abstract: Please proportionally present background, purpose, methods, results, and conclusion.
  2. Introduction: The ‘Introduction’ section is well-written and nicely presented, with a good balance of descriptive text and information about biological pathways in mood disorders. Even though the authors decided to take a narrow view of the cognitive impairment related to biological dysregulation in psychiatric disorders, I believe that a deeper examination of the mechanisms underlying impairments in emotional learning, memory and how this ability, together with deficient inhibitory control, are core factors in different psychopathologies, such as bipolar disorder, would provide a useful background. Interestingly, results from a recent review (https://doi.org/10.1016/j.brat.2021.103963) outlined typical dysfunctional behaviors, such as deficit in action control and motor inhibition, that are associated with psychopathological and psychiatric conditions, which are characterized by severe impulsivity problems that can determine significant impairment or distress (due to poor regulation and capacity of control, which can be intensified in the presence of emotional stimuli). Accordingly, another recent review (https://doi.org/10.3390/biomedicines10030627), focused on pathological mechanisms underlying altered emotion perception, which is significantly impaired in brain-damaged patients, are related to amygdala and superior temporal sulcus dysfunctions. Finally, I also believe that a recent perspective manuscript on the metabolic pathways (https://doi.org/10.17219/acem/139572) involved in the pathogenesis of a wide range of diseases might be of interest. Moreover, if they deem it appropriate, authors can also check additional studies that have focused on searching for biomarkers, developing precision diagnostics, and underlying processes causing impairments in social cognition and social functioning, associated with various psychiatric, neurological and neurodegenerative illnesses (https://doi.org/10.3390/life11121365; https://doi.org/3390/biomedicines9040403; https://doi.org/10.3390/biomedicines10010076; https://doi.org/10.3390/biomedicines9050517; https://doi.org/10.3390/biomedicines9070734).
  3. Methods: Please name two authors who conducted search and the third author with initials.
  4. Results: In my opinion, this section is well organized; however, as the authors have pointed out, very few studies were included in the review/meta-analysis. Therefore, to ensure in-depth understanding and replicability of the findings, I suggest better describing in detail the few studies reported in this review, by providing a detailed description of the hypothesis, the proteomics strategies used to study BD, the results and their implications. Also, in my opinion, it is necessary for the authors to present their findings using summary tables.
  5. Discussion: In my opinion, this research article would be more compelling and useful to a broad readership if the authors moved beyond and discussed theoretical and methodological avenues in need of refinement, using this evidence to suggest a path forward. In this regard, I believe that it would have been essential to investigate the neurobiology of mental disorders in humans, and examine the implementation of new therapeutic techniques, such as Non-invasive brain stimulation, that operate to ameliorate the symptoms of mental and neurological disorders. In this regard, I would suggest evidence from recent studies that have examined NIBS efficacy: results from an innovative recent study showed that inhibition of the dorsolateral prefrontal cortex (DLPFC) with rTMS after emotional memory reactivation disrupts physiological responding to learned fear, highlighting the role of this area in the neural network of fear memories in humans (https://doi.org/10.1016/j.cub.2020.06.091).
  6. Discussion: Following the previous point raised, in order to provide a more thorough background on this topic, I would also suggest, a recent review (https://doi.org/10.1016/j.neubiorev.2021.04.036) that described the potential and effectiveness of non-invasive brain stimulation (NIBS) to interfere and modulate the abnormal activity of neural circuits (i.e., amygdala-mPFC-hippocampus) involved in the acquisition and consolidation of fear memories, which are altered in many mood psychiatric disorders (i.e., bipolar disorder, anxiety disorder, specific phobias, post-traumatic stress disorder or depression). Similarly, another recent study illustrated the therapeutic potential of NIBS as a valid alternative in the treatment of abnormally persistent fear memories that characterized those patients with anxiety disorders that do not respond to psychotherapy and/or drug treatments (https://doi.org/10.1016/j.jad.2021.02.076). In addition to the previously mentioned literature, authors can also see these additional studies that have focused on this topic (https://doi.org/10.2174/1871527313666141130224431; https://doi.org/10.1016/j.neubiorev.2018.05.015). These findings highlight how NIBS and are a valuable tool in research and has potential diagnostic and therapeutic applications for many mood psychiatry disorders, including bipolar disorder, depression or anxiety.
  7. In my opinion, I think the ‘Conclusions’ paragraph would benefit from some thoughts as well as in-depth considerations by the authors because as it stands, it is very descriptive but not enough theoretical as a discussion should be. Authors should make an effort, to explain the theoretical implication and the translational application of their research.
  8. In according with the previous comment, I would ask the authors to include a ‘Limitations and future directions’ section before the end of the manuscript, in which authors can describe in detail and report all the technical issues brought to the surface.
  9. Figures: I suggest modifying the Figure 5 for clarity because, as it stands, the readers may have difficulty comprehending it. Also, all figures and tables should be numbered following their number of appearance and should have a short explanatory title and caption within the main text, to ensure consistency with the captions listed at the end of the manuscript.
  10. References: According to the Journal’s guidelines, authors should consider revising the bibliography, as there are several incorrect citations. Indeed, according to the Journal’s guidelines, they should provide the abbreviated journal name in italics, the year of publication in bold, the volume number in italics for all the references. Also, they should have provided the DOI number for each reference.

Overall, the manuscript contains five figures, two tables and 98 references. I believe that the manuscript may carry important value investigating the potential of mass spectrometry clinical proteomic studies in the future search of biomarkers for BD. I hope that, after these careful revisions, the manuscript can meet the Journal’s high standards for publication. I am available for a new round of revision of this review.

Best regards,

Reviewer

Author Response

Next we present reviewer comments followed by our answers after the bullet point:

Rodrigues and colleagues investigated in the present review entitled ‘Systematic review and meta-analysis on MS-based proteomics applied to human peripheral fluids to assess potential biomarkers of bipolar disorder’, the current status of knowledge of the efficacy of mass spectrometry (MS) proteomics applied to human peripheral fluids to assess bipolar disorder (BD) biomarkers and identify relevant networks of biological pathways. For this purpose, the authors identified fourteen articles, that allowed the identification of 266 differently expressed proteins, complement and coagulation cascades, lipid and cholesterol metabolism, and focal adhesion as the main enriched biological pathways. Although the proven ability of MS proteomics to characterize BD, there were several confounding factors contributing to the heterogeneity of the findings. The authors concluded by stating that in the future, it would be necessary to integrate omics with bioinformatics tools to provide a comprehensive understanding of proteome alterations, biomarkers of BD, and to contribute to individualized prognosis and stratification strategies.

The main strength of this original review article is that it addresses an interesting and timely question, investigating the potential of mass spectrometry clinical proteomic studies in the future search of biomarkers for BD. In general, I think the idea of this article is really interesting and the authors’ fascinating observations on this timely topic may be of interest to the readers of the International Journal of Molecular Sciences.

  • We appreciate the positive opinion of the reviewer to our work.

However, some comments, as well as some crucial evidence that should be included to support the author’s argumentation, needed to be addressed to improve the quality of the manuscript, its adequacy, and its readability before its publication in the present form, in particular reshaping parts of the Introduction and Discussion sections by adding more evidence and theoretical constructs.

Please consider the following comments:

  1. Abstract: Please proportionally present background, purpose, methods, results, and conclusion.
  • we included more information on the background, and we believe that the abstract is now more balanced, although we have no indication that we should divide by section.

  1. Introduction: The ‘Introduction’ section is well-written and nicely presented, with a good balance of descriptive text and information about biological pathways in mood disorders. Even though the authors decided to take a narrow view of the cognitive impairment related to biological dysregulation in psychiatric disorders, I believe that a deeper examination of the mechanisms underlying impairments in emotional learning, memory and how this ability, together with deficient inhibitory control, are core factors in different psychopathologies, such as bipolar disorder, would provide a useful background. Interestingly, results from a recent review (https://doi.org/10.1016/j.brat.2021.103963) outlined typical dysfunctional behaviors, such as deficit in action control and motor inhibition, that are associated with psychopathological and psychiatric conditions, which are characterized by severe impulsivity problems that can determine significant impairment or distress (due to poor regulation and capacity of control, which can be intensified in the presence of emotional stimuli). Accordingly, another recent review (https://doi.org/10.3390/biomedicines10030627), focused on pathological mechanisms underlying altered emotion perception, which is significantly impaired in brain-damaged patients, are related to amygdala and superior temporal sulcus dysfunctions. Finally, I also believe that a recent perspective manuscript on the metabolic pathways (https://doi.org/10.17219/acem/139572) involved in the pathogenesis of a wide range of diseases might be of interest. Moreover, if they deem it appropriate, authors can also check additional studies that have focused on searching for biomarkers, developing precision diagnostics, and underlying processes causing impairments in social cognition and social functioning, associated with various psychiatric, neurological and neurodegenerative illnesses (https://doi.org/10.3390/life11121365; https://doi.org/3390/biomedicines9040403; https://doi.org/10.3390/biomedicines10010076; https://doi.org/10.3390/biomedicines9050517; https://doi.org/10.3390/biomedicines9070734).

  • The authors appreciate all suggestions to improve the introduction. However, we believe that the points raised by the reviewer would lead our path in a different direction. The authors wanted to briefly outline the disorder and focus then on mass spectrometry proteomics screening on peripheral fluids. Not addressing other protein and metabolites identified by other approaches. Also, the second reviewer suggested a reduction of the introduction. Therefore we performed some text shortening.

  1. Methods: Please name two authors who conducted search and the third author with initials.
  • We indicated this information in the current resubmission.

  1. Results: In my opinion, this section is well organized; however, as the authors have pointed out, very few studies were included in the review/meta-analysis. Therefore, to ensure in-depth understanding and replicability of the findings, I suggest better describing in detail the few studies reported in this review, by providing a detailed description of the hypothesis, the proteomics strategies used to study BD, the results and their implications. Also, in my opinion, it is necessary for the authors to present their findings using summary tables.
  • For a meta-analysis we believe that we gave an explanation above the average and all data is present in comprehensive tables. We believe the document is already too long explaining the different studies and an even more detailed description could become too heavy and remove focus from the meta-analysis.

  1. Discussion: In my opinion, this research article would be more compelling and useful to a broad readership if the authors moved beyond and discussed theoretical and methodological avenues in need of refinement, using this evidence to suggest a path forward. In this regard, I believe that it would have been essential to investigate the neurobiology of mental disorders in humans, and examine the implementation of new therapeutic techniques, such as Non-invasive brain stimulation, that operate to ameliorate the symptoms of mental and neurological disorders. In this regard, I would suggest evidence from recent studies that have examined NIBS efficacy: results from an innovative recent study showed that inhibition of the dorsolateral prefrontal cortex (DLPFC) with rTMS after emotional memory reactivation disrupts physiological responding to learned fear, highlighting the role of this area in the neural network of fear memories in humans (https://doi.org/10.1016/j.cub.2020.06.091).
  • This is an interesting field of research but would deviate from the MS proteomics-based identification of biomarkers in peripheral fluids which was the main objective of this systematic review.

  1. Discussion: Following the previous point raised, in order to provide a more thorough background on this topic, I would also suggest, a recent review (https://doi.org/10.1016/j.neubiorev.2021.04.036) that described the potential and effectiveness of non-invasive brain stimulation (NIBS) to interfere and modulate the abnormal activity of neural circuits (i.e., amygdala-mPFC-hippocampus) involved in the acquisition and consolidation of fear memories, which are altered in many mood psychiatric disorders (i.e., bipolar disorder, anxiety disorder, specific phobias, post-traumatic stress disorder or depression). Similarly, another recent study illustrated the therapeutic potential of NIBS as a valid alternative in the treatment of abnormally persistent fear memories that characterized those patients with anxiety disorders that do not respond to psychotherapy and/or drug treatments (https://doi.org/10.1016/j.jad.2021.02.076). In addition to the previously mentioned literature, authors can also see these additional studies that have focused on this topic (https://doi.org/10.2174/1871527313666141130224431; https://doi.org/10.1016/j.neubiorev.2018.05.015). These findings highlight how NIBS and are a valuable tool in research and has potential diagnostic and therapeutic applications for many mood psychiatry disorders, including bipolar disorder, depression or anxiety.
  • Again, these studies are very interesting but would deviate from the main purpose of the study focused on MS proteomics-based identification of biomarkers in peripheral fluids

  1. In my opinion, I think the ‘Conclusions’ paragraph would benefit from some thoughts as well as in-depth considerations by the authors because as it stands, it is very descriptive but not enough theoretical as a discussion should be. Authors should make an effort, to explain the theoretical implication and the translational application of their research.
  • We appreciate the reviewer comments. We already indicate how next studies could be performed “Further studies with broad samples and validation cohorts, longitudinal designs with patients in early phases of BD, and integrating proteomics results with other omics data (phenomics, genomics, metabolomics, connectomics) could provide additional information about differentially expressed proteins in selected biological pathways. The comprehensive information produced could harbor proteomic biomarkers of BD, contributing to individualized prognosis and stratification strategies, besides aiding in the differential diagnosis.” We understand that the conclusion is quite descriptive and exactly for that reason we would like to refrain from overstating the results. Potentially, an opinion article or a letter could give a broader opinion on future directions.

  1. In according with the previous comment, I would ask the authors to include a ‘Limitations and future directions’ section before the end of the manuscript, in which authors can describe in detail and report all the technical issues brought to the surface.
  • this was included in the original submitted manuscript “STRENGTHS AND LIMITATIONS” and “DIRECTIONS FOR FUTURE RESEARCH”

  1. Figures: I suggest modifying the Figure 5 for clarity because, as it stands, the readers may have difficulty comprehending it. Also, all figures and tables should be numbered following their number of appearance and should have a short explanatory title and caption within the main text, to ensure consistency with the captions listed at the end of the manuscript.
  • Figure 5 was shortened by one element (focal adhesion, which is now a supplementary figure S3) so we could increase each figure element size and the resolution was also increased.

  1. References: According to the Journal’s guidelines, authors should consider revising the bibliography, as there are several incorrect citations. Indeed, according to the Journal’s guidelines, they should provide the abbreviated journal name in italics, the year of publication in bold, the volume number in italics for all the references. Also, they should have provided the DOI number for each reference.
  • We changed the document accordingly

Overall, the manuscript contains five figures, two tables and 98 references. I believe that the manuscript may carry important value investigating the potential of mass spectrometry clinical proteomic studies in the future search of biomarkers for BD. I hope that, after these careful revisions, the manuscript can meet the Journal’s high standards for publication. I am available for a new round of revision of this review.

Reviewer 2 Report

Dear Authors,

I read with great interest your manuscript. However, I have several comments:

Please revise once again the entire manuscript since I identified free spaces and attention to the style. 

The Introduction is extremely long. Can it be shortened to an extent?

Please maximize the figure since the compilation of pathways presented are too small and hard to understand.

Supplementary material, in spite, tables did not fit on the page.

Although it seems complete, how does this manuscript differ from the others in the literature?!

Kind regards,

The Reviewer

Author Response

Please find below the reviewer's comments followed by our answer after the bullet point:

I read with great interest your manuscript. However, I have several comments:

Please revise once again the entire manuscript since I identified free spaces and attention to the style.

  • The document was revised accordingly

The Introduction is extremely long. Can it be shortened to an extent?

  • we revised the introduction and reduced some of its content. However, the other reviewer suggested that we should add even more information. Overall, we believe that this is the necessary background to address the objective of the manuscript.

Please maximize the figure since the compilation of pathways presented are too small and hard to understand.

  •  Figure 5 was shortened by one element (focal adhesion, which is now a supplementary figure) so we could increase each figure element size and the resolution was also increased.

Supplementary material, in spite, tables did not fit on the page.

  • The Table was corrected

Although it seems complete, how does this manuscript differ from the others in the literature?!

  • To our knowledge there is no other meta-analysis performed on mass-spectrometry based proteomics approaches performed on peripheral fluids to identify potential biomarkers for bipolar disorder to be used in the clinical practice. This manuscript is also very important to highlight that most reports are not consistent in data reporting and clinical information that could lead to such biomarker validation in larger cohorts.

Round 2

Reviewer 1 Report

2 May 2022

Regarding the 2nd review of manuscript “Systematic review and meta-analysis on MS-based proteomics applied to human peripheral fluids to assess potential biomarkers of bipolar disorder” by Rodrigues JE et al., submitted to International Journal of Molecular Sciences

Manuscript ID: ijms-1667369

Dear Authors,

In this study by Rodrigues and colleagues, authors aimed to explore the current status of knowledge regarding the efficacy of mass spectrometry (MS) proteomics applied to human peripheral fluids to assess bipolar disorder (BD) biomarkers and to identify relevant networks of biological pathways.

I appreciated the Authors' answers to the points that I raised in the first round of review, as well as their clarifications of some of my concerns. However, despite my suggestions to provide more information, by adding some crucial studies that could have allowed to enrich and complete the theoretical framework, the authors have stated that they do not feel the necessity to add more evidence. Personally, I still believe that add findings from the previously suggested studies that have focused on biological dysregulation in psychiatric disorders, and how these impact cognitive abilities in psychopathologies, would help deepen the subject of this manuscript (https://doi.org/10.3390/biomedicines9070734; https://doi.org/10.3390/ijms23042294; https://doi.org/10.3390/biomedicines10030627; https://doi.org/10.1016/j.brat.2021.103963).

I hope authors would carefully consider my suggestions.

I am always available for other reviews of such interesting and important articles.

Thank you for your work,

Reviewer

Author Response

Dear Reviewer

We appreciate your comments, and we are happy to know that our revision clarified the document. We carefully read the manuscripts which you mentioned, we added a sentence and included one reference. We still believe that the other works which were suggested, although very interesting and important in their field, considerably deviate from the biomarker analysis in peripheral fluids by mass spectrometry-based proteomics and the potential origin of such biomarkers.

We hope that we have improved the document and explained why we consider that we would deviate the audience focus from the goal of our manuscript.

Kind regards

Reviewer 2 Report

Dear Authors,

Congratulations on the hard work reviewing our manuscript. Indeed. It has been significantly improved. Based on this consideration, I recommend acceptance in the current form.

Kind regards,

The Reviewer

Author Response

Dear Reviewer

thanks for your comments and your contribution to a better manuscript.